# DATATYPE TAGGING AND PROMPT ALIGNMENT: A RECIPE FOR BOOSTING LLMS ON ALGORITHMIC TASKS

## ABSTRACT

This paper contributes toward strengthening the bridge between LLMs as programmers and classical ideas in programming languages (PL). Specifically, we show that aligning prompts with *typed programs* enables even small models to reliably emit one-line Python code. We present a simple yet effective recipe consisting of three key ingredients: (i) inline datatype tagging for prompt and code; (ii) a fine-tuned dual-head GPT-2-small with an auxiliary span probe over the prompt; and (iii) a fixed decoder that enforces a finite-state grammar, validates AST shape, and repairs outputs deterministically. On a stratified GPT-4o based dataset that covers primitives such as add, subtract, max, min, and sort, the decoder alone raises execution accuracy by over 40% (from 0.58 to 0.82)! For counting and repeated addition, prompts map deterministically to single expressions (for example, s.count('r') and sum([1]*100)), yielding near-zero errors within coverage. Our approach runs on a single GPU, and presents a proof-of-concept on how "datatype-aware tokenization" and "grammar-first decoding," among other ideas inspired by PL, improve reliability, coverage, and quality at low cost.

## 1 INTRODUCTION

Users often ask AI assistants to carry out small computational tasks such as basic arithmetic, process sequences (e.g., adding, computing max, etc.), or other algorithmic tidbits. Most commonly, for such tasks the user prompts in plain English (or another language), which forces the model to infer *datatypes* and reconstruct a single line program while dealing with many different phrasings of the same task. That extra work raises uncertainty and invites hallucination; unsurprisingly, arithmetic and short program tasks remain brittle for general purpose models without imposing additional structure (Cobbe et al., 2021; Ji et al., 2023). Reasoning style prompting helps, yet it still keeps planning in text (Wei et al., 2022; Wang et al., 2022; Drozdov et al., 2022). Program or tool aided prompting offloads execution (Gao et al., 2022; Chen et al., 2022; Yao et al., 2023; Schick et al., 2023), but it typically assumes a strong model that already emits clean code or well formed structured calls.

We take a different stance: we treat these requests as typed program emission. We make datatypes explicit in the input, we constrain decoding to a small set of legal shapes, and we execute inside a sandbox. More precisely, our recipe comprises: (i) inline datatype tags; (ii) a small dual head student with a code LM head and an auxiliary head that highlights important tokens in the prompt; (iii) a fixed decoder with a DFA, an AST check, and a canonicalizer. Together these ideas make a small model dependable on this narrow set of algorithmic tasks and greatly improve accuracy without increasing model size. We illustrate our plan through a concrete proof-of-concept, for which we now formally express the associated research question and our hypotheses.

> **Main question.** *Can a small LM reliably emit one line Python for basic algorithmic prompts if we make datatypes explicit and restrict generation to tokens allowed by a small grammar?*
>
> **Our Hypotheses.**
> *(H1) Inline datatype tags align prompt and code tokens and narrow the next token choices.*
> *(H2) A small token level tagger over the prompt stabilizes training and helps the model locate numbers and list boundaries.*
> *(H3) Grammar constrained decoding with a DFA during generation, plus an AST check and a deterministic repair step, boosts accuracy.*

## 1.1 MAIN CONTRIBUTIONS

In light of the above, we are now ready to summarize our key conceptual and practical contributions.

1. **A simple paradigm.** We propose to treat short algorithmic requests as typed program emission. We make datatypes visible in the prompt with inline tags and we constrain decoding with a small grammar gate (a deterministic finite automaton, DFA) and an abstract syntax tree (AST) check. Importantly, this design moves structure into the interface and into decoding and gives an auditable path from prompt to one line of code. We also formalize the target as a regular language and give a DFA for it, prove soundness of the DFA plus AST checks, and show completeness within our covered tasks for a deterministic canonicalizer (a rule-based repair that maps a prompt to one legal line when needed). We provide simple time bounds and safety invariants. See Theoretical properties (subsection 3.1) for details. Figure 1 and Figure 2 illustrate the idea.

2. **A learning recipe that aligns prompts with code.** We finetune a small two head GPT-2 model on tagged sequences. The main head learns to generate tagged code. A light auxiliary head learns to highlight numbers and list boundaries in the prompt. The auxiliary head is used only during training. This recipe narrows the choices the model must consider and improves training stability while keeping the model small. Figure 1 shows the training flow.

3. **Constrained decoding with deterministic repair.** During generation the DFA blocks invalid next tokens and the AST check enforces the expected shape. If a string is not acceptable a deterministic canonicalizer rebuilds one legal line from the prompt rather than sampling again. Figure 3 shows streaming under the grammar gate, Figure 4 shows the minimal list DFA, and Figure 5 shows canonical rewrites from tagged prompts to one line Python.

Based on the above contributions, we envision a path to eventual industrial deployment. But more modestly, toward validation of our ideas, we create "gold code"[1] locally and validate it with parsing and safe execution. A stronger teacher model writes prompts that preserve numerals and list entries. On a stratified set of GPT-4o prompts for add, sub, max, min, and sort, guarded decoding alone raises execution accuracy by over 40%, from 0.58 to 0.82. Tags reduce malformed outputs and the auxiliary head improves training stability. We also sketch an industrial router that sends algorithmic requests to this typed microservice and routes other traffic to a general model pool (Figure 6).

## 1.2 RELATED WORK

**Reasoning by prompting.** Chain of Thought, Self Consistency, and Least to Most improve accuracy by sampling and by decomposing rationales (Wei et al., 2022; Wang et al., 2022; Drozdov et al., 2022). Planning remains in text, which leaves output structure unconstrained. Our approach moves structure into the interface by exposing types and by constraining the decode.

**Program and tool use.** Program and tool aided methods execute generated code or invoke tools (Gao et al., 2022; Chen et al., 2022; Press et al., 2022; Yao et al., 2023; Schick et al., 2023). Industry practice mirrors this through function calling and structured outputs that enforce JSON schemas at inference time, which makes downstream integration type safe (OpenAI, 2023; 2024). Large production models highlight structured reasoning and tool readiness in technical reports such as PaLM 2 (Anil et al., 2023). We aim for similar reliability with a much smaller model by aligning tokens and by enforcing a compact grammar.

**Constrained generation.** Several frameworks enforce output shape during decoding, including PICARD for Text to SQL (Scholak et al., 2021), Synchromesh for code (Poesia et al., 2022), and token level schema guidance in Outlines and DOMINO (Willard & Louf, 2023; Beurer-Kellner et al., 2024). LMQL expresses constraints and control flow as a query language that prunes invalid continuations and reduces cost (Beurer-Kellner et al., 2023). We follow the same spirit of structured decoding while adding input side typing, span supervision, and a deterministic repair path that together make a small model reliable on our task (Poesia et al., 2022; Scholak et al., 2021; Willard & Louf, 2023; Beurer-Kellner et al., 2024).

---

[1]"Gold code" is the gold-standard one line Python program that serves as the reference for accuracy.

**Algorithm 1** TAG_TEXT inserts datatype tags and guarantees exact recovery.

```python
def tag_text(s: str) -> str:
    s = normalize_ascii(s) # keep quotes; normalize hyphens, spaces
    toks = lex(s) # words, digits, quotes, brackets, commas
    out = []
    for t in toks:
        if is_int_literal(t): out += [INT, t]
        elif is_float_literal(t): out += [FLOAT, t]
        elif is_bool_literal(t): out += [BOOL, t]
        elif is_quoted_string(t): out += [STR, t]
        elif t == "[": out += [LIST, "["]
        elif t == "]": out += [LIST, "]"]
        elif t == "(": out += [TUPLE, "("]
        elif t == ")": out += [TUPLE, ")"]
        else: out += [t]
    return "".join(out)
```

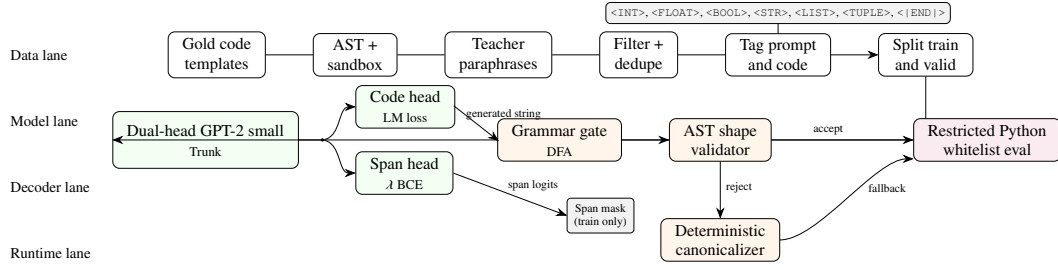

Figure 1: **Full pipeline from data to execution.** We first create gold single line programs and check them with AST parsing and safe evaluation. A stronger teacher writes matching prompts that keep numerals and list entries unchanged. We clean and deduplicate the pairs, add inline type tags to the prompt and the code, and build the training and validation splits. The model reads the tagged prompt, the end token, and the tagged code. The code head learns with a causal language modeling loss, and a small span head learns to mark the argument tokens in the prompt. At inference a deterministic finite automaton restricts decoding to legal shapes, and an AST validator checks the result. If the output is not acceptable a canonicalizer rebuilds a single legal line from the prompt. The final expression runs in a restricted Python sandbox that allows basic arithmetic and the functions `max`, `min`, and `sorted`.

**Code LMs and verification.** Codex, AlphaCode, and Code Llama increase their pass rates through scale and verification (Chen et al., 2021; Li et al., 2022; Rozière et al., 2023). Our goal is orthogonal: we recover reliability with low compute by shrinking the search space and by validating AST shape.

**Routing, mixtures, and guardrails.** Mixture of Experts activates sparse experts inside a model (Shazeer et al., 2017; Lepikhin et al., 2020; Fedus et al., 2021; Du et al., 2022). System routers and cascades choose among models or tools for quality and cost (Chen et al., 2023; Ding et al., 2024; Wang et al., 2024; Dohan et al., 2022), and confidence aware early exit reduces latency when uncertainty is low (Schuster et al., 2022). Guardrails in production stacks such as NeMo Guardrails declaratively enforce policies and schemas around LLMs, and our grammar and AST checks play a similar role at decode time (NVIDIA, 2023).

## 2 PROBLEM SETUP AND EXPERIMENTAL BACKGROUND

Our goal is to present a simple and general recipe that makes types explicit in the prompt, constrains decoding with a small grammar, and runs the result in a sandbox. To make the ideas concrete we use a compact set of single line algorithm expressions that provide clean ground truth, an exact grammar, and a safe runtime. These examples are meant to serve as testbeds rather than a limit on scope. Indeed, the same principles apply whenever outputs fit a small schema.

We instantiate the recipe with five simple one-line algorithmic expressions over integers and lists:

add is $a+b$, sub is $a-b$, max is $\max([x_1,\ldots,x_n])$, min is $\min([x_1,\ldots,x_n])$, sort is $\mathrm{sorted}([x_1,\ldots,x_n])$,

with $a, b, x_i \in \mathbb{Z}$ and $n \geq 1$. The system must emit a single Python expression. Multi line code, imports, side effects, and calls outside a small allowed set are not permitted. The allowed set is integer arithmetic and `max`, `min`, and `sorted`. We also analyze two auxiliary families, character counting and repeated addition, that map deterministically to single lines.

We make structure explicit on the input and the output. Deterministic mappers $T$ for the prompt and $U$ for the code insert inline datatype tags so numerals, strings, booleans, and container boundaries are visible to the model (Algorithm 1). A detagger $D$ removes tags with a byte for byte guarantee, so $D(T(s)) = s$ and $D(U(y)) = y$. Decoding is constrained by a compact grammar that we compile to a DFA. Only strings accepted by the DFA are sent to an AST shape validator, and only validated expressions are executed in a sandbox. We report execution accuracy

$$\mathrm{ExecAcc} = \frac{1}{N} \sum_i \mathbf{1}\big[\mathrm{eval}(e_i) = \mathrm{eval}(e_i^\star)\big]$$

after detagging. We also track diagnostics that include the DFA and AST pass rate, the fraction of malformed outputs that are blocked, an error breakdown, and code segment cross entropy. Unless noted, we use a stratified suite of 60 GPT 4o prompts that preserve numerals and list contents and cover *add*, *sub*, *max*, *min*, and *sort*.

*Dual head objective.* We concatenate inputs as $x \parallel \langle\mathrm{END}\rangle \parallel y$, where $x$ is the tagged prompt and $y$ is the tagged code. The main head is a causal LM trained only on the code segment,

$$\mathcal{L}_{\mathrm{LM}} = - \sum_t \log p_\theta(y_t \mid x, \langle\mathrm{END}\rangle, y_{<t}),$$

with tokens in $x$ and at `<|END|>` masked out. A span head learns a binary mask over the prompt that highlights digits and container punctuation,

$$\mathcal{L}_{\mathrm{span}} = \frac{\sum_i a_i p_i \, \mathrm{BCE}\big(\sigma(s_i), m_i\big)}{\sum_i a_i p_i}, \qquad \mathcal{L} = \mathcal{L}_{\mathrm{LM}} + \lambda \, \mathcal{L}_{\mathrm{span}}, \ \lambda \in [0.25, 1.0].$$

At inference we use only the code head, and the span head is diagnostic.

*Grammar constrained decoding and repair.* The grammar

$$S \to \mathrm{Add} \mid \mathrm{Sub} \mid \mathrm{Max} \mid \mathrm{Min} \mid \mathrm{Sort}, \quad \mathrm{Add} \to \texttt{INT} + \texttt{INT}, \ \mathrm{Sub} \to \texttt{INT} - \texttt{INT},$$

$$\mathrm{Max} \to \texttt{max([ELTS])}, \ \mathrm{Min} \to \texttt{min([ELTS])}, \ \mathrm{Sort} \to \texttt{sorted([ELTS])},$$

uses `INT=` `[-]?\d+` where the minus is the ASCII hyphen, and ELTS is a comma separated list with at least one `INT` and no trailing comma. We compile this to a DFA with a sink for illegal steps. The list subautomaton alternates between "expect int" and "expect comma or ]", which enforces at least one element and no trailing comma. Inference is greedy under the DFA. We emit a token only if a valid transition exists, cap the length, stop on EOS or newline or `\endtok`, normalize U+2212 to $-$, take the first ASCII line, and detag. We then parse to an AST and enforce exact shape. If checks fail, a deterministic canonicalizer reconstructs a legal expression by intent detection with small synonym sets, number and sign normalization, enumeration extraction, and order fixes.

## 3 METHOD, DATASETS, AND RESULTS

**Model and training.** We fine tune GPT-2 small with a tokenizer augmented by $\{$`<INT>`, `<FLOAT>`, `<BOOL>`, `<STR>`, `<LIST>`, `<TUPLE>`, `<|END|>`$\}$. Each example is `tagged_prompt` $\parallel$ `<|END|>` $\parallel$ `tagged_code`. The main head is a causal LM trained only on the code segment; a light span head predicts a per token mask on the prompt (digits and container punctuation). At inference only the code head is used. Training runs for 1–3 epochs (learning rate $2 \times 10^{-5}$, batch 8/16, max length 256). Base and auxiliary heads are saved separately to avoid tied weight issues.

**Data construction.** For each skill, gold single line code is synthesized locally, validated with `ast.parse` and safe execution, and paired with prompts from a stronger teacher (GPT-4o or -4o

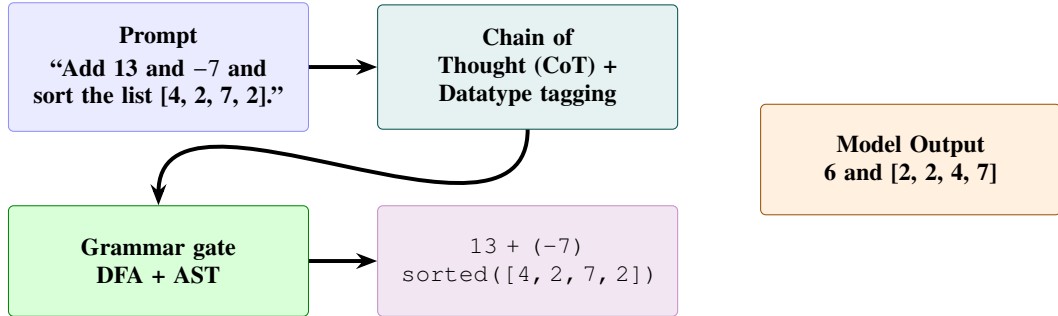

Figure 2: **Pairing datatype tagging with CoT.** A plain prompt often leads to long chain of thought text, extra words, and sometimes malformed code before the answer. The same prompt with datatype tagging and a simple grammar gate will give an immediate, clean result. CoT can still guide the plan, but tags expose numbers and list boundaries and the DFA plus AST check allow only a single legal Python line. In the example we ask to add 13 and -7 and then sort the list [4, 2, 7, 2]. The system produces `13 + (-7)` and `sorted([4,2,7,2])` and returns 6 and [2, 2, 4, 7] right away.

mini) instructed to preserve numerals and list contents. Rows store {`source`, `skill`, `prompt`, `code`, `tagged_prompt`, `tagged_code`} and are deduplicated by normalized prompt and skill. Evaluation uses a stratified held out suite of 60 GPT-4o prompts covering *add/sub/max/min/sort*; counting and repeated addition are analyzed separately.

**Findings.**

- *Datatype tags help.* Training two identical students that differ only in tagging shows lower code segment cross entropy and fewer malformed greedy generations with tags, especially fewer bracket and operand order errors (Figure 2). Tags align prompt and code vocabularies and shrink the search space.

- *Constrained decoding drives accuracy.* Starting from the tagged student, we compare greedy decoding; DFA with an AST check; and DFA with an AST check plus a deterministic canonicalizer. On the 60 prompt suite, execution accuracy rises from **0.58** to **0.82** without any change to the model or the data. The largest gains are in subtraction and sorting, where the validator and canonicalizer correct operand order and bracketing (Figures 3–5). The DFA also removes a long tail of malformed strings and improves accept rates.

- *Span probe stabilizes training.* Adding the span head yields more stable optimization across seeds and masks that clearly localize numerals and container boundaries. With grammar, AST, and the canonicalizer at inference, end accuracy is comparable or slightly higher, so the probe mainly aids stability and diagnostics rather than test time capacity.

- *Deterministic families need no model.* For character counting and repeated addition, a canonicalizer maps prompts directly to `s.count(c)` and `sum([x]*n)` after synonym and number normalization. Within stated coverage (ASCII and bounded lengths), errors are near zero, which removes model calls and an entire class of failures.

## 3.1 THEORETICAL PROPERTIES OF THE METHOD

Let $\mathcal{K} = \{\texttt{max}, \texttt{min}, \texttt{sorted}\}$ and define

$$\texttt{INT} := \texttt{[-]?} \textbf{+} \text{ (ASCII minus sign)}, \qquad \texttt{ELTS} := \texttt{INT(,INT)}^*,$$

so the target language is

$$\mathcal{L} = \{\texttt{INT+INT}, \texttt{INT-INT}\} \cup \bigcup_{k \in \mathcal{K}} \{\, k(\texttt{[ELTS])} \,\}.$$

Because ELTS is flat (no nesting), each subset is regular and therefore $\mathcal{L}$ is regular; a deterministic finite automaton exists. Our compiled DFA uses a constant number of states (on the order of a few dozen), performs $O(1)$ transitions per token, and ignores whitespace between tokens.

*Soundness.* If a token stream $y$ is accepted by the DFA and passes the AST shape check, then $D(y)$ parses to either $\texttt{BinOp}(+/-)$ on two integer constants or a whitelisted call in $\mathcal{K}$ with a list of integer constants. Evaluation in the sandbox is pure and without side effects, so the returned value matches the denotational semantics of $\mathcal{L}$.

*Completeness within coverage.* The deterministic canonicalizer $C$ maps any in-coverage prompt (add, subtract, max, min, or sort with recoverable numerals) to some $C(p) \in \mathcal{L}$ that always passes the DFA and AST checks; when its preconditions do not hold, the system abstains.

*Determinism and invariances.* Greedy decoding under DFA gating with a fixed tokenizer is deterministic, and so is $C$. Normalizing the Unicode minus (U+2212 → −) and benign whitespace edits do not change acceptance.

*Complexity and safety.* With $m$ generated tokens and $n$ list elements, DFA guarded decoding runs in $O(m)$ time and $O(1)$ memory; AST validation is $O(m)$; sandbox evaluation is $O(1)$ for add and subtract, $O(n)$ for $\texttt{max}$ and $\texttt{min}$, and $O(n \log n)$ for $\texttt{sorted}$; $C$ runs in $O(|p|)$. Grammar and AST checks forbid attribute access, arbitrary calls, comprehensions, f strings, and imports; only $\{+, -, \texttt{max}, \texttt{min}, \texttt{sorted}\}$ are permitted on integer payloads. Together with detagging idempotence $D(T(s)) = s$ and a deny by default sandbox, any returned string is either in $\mathcal{L}$ and safe to execute, or the system abstains.

## 4 DISCUSSION AND IMPLICATIONS

Datatype tags align the token stream with the program space, so the model does not need to guess that digits are integers or that brackets mark containers. A small DFA turns decoding into a short and safe search over a few legal shapes, and a deterministic canonicalizer provides a reliable fallback. Because each part is small, the pipeline is easy to audit and test. This mirrors guided generation frameworks that prune invalid continuations during decoding (Scholak et al., 2021; Poesia et al., 2022; Willard & Louf, 2023; Beurer-Kellner et al., 2024; 2023). As a systems pattern, typed prompting with grammar first decoding complements function calling and structured outputs by moving structure into the decode itself (OpenAI, 2023; 2024). It also fits cascades and budget aware routers, where we accept when grammar and AST checks pass and otherwise abstain and defer to a larger model (Dohan et al., 2022; Chen et al., 2023; Ding et al., 2024; Wang et al., 2024).

The same recipe extends to multimodal generation. Diffusion systems already expose typed control channels such as edges, depth, keypoints, boxes, and masks (Zhang & Agrawala, 2023; Mou et al., 2023). Attention level methods and compositional or classifier free guidance improve semantic faithfulness (Chefer et al., 2023; Hertz et al., 2022; Liu et al., 2022; Ho & Salimans, 2022). Our tags can act as a front end schema, for example a JSON prompt with $\texttt{objects=[\{class,bbox,color\}]}$ and $\texttt{style=\{palette,lighting\}}$, which routes fields to the right control adapters and softly enforces counts and placements. For video, tags for shot list, duration, camera motion, and trajectories can compile into per frame control streams for text to video models (Ho et al., 2022; Wang et al., 2023; Guo et al., 2023; Singer et al., 2022).

Typed prompting is also natural for graph structured models, where schemas for node and edge types and attributes are explicit. Recent work shows that pretrained GNNs can be steered with small structured hints (Sun et al., 2023; Lee et al., 2024; Wu et al., 2023). A grammar first interface can validate or synthesize a graph DSL before a GNN or a solver, which aligns with neural algorithmic reasoning when the target computation has a known shape (Veličković & Blundell, 2021). There is a path at pretraining time as well. A tag aware continued pretraining step in the spirit of T5 span corruption with reserved whole tag symbols, a token level span probe, and a small grammar prediction head could bake these inductive biases into larger instruction models (Raffel et al., 2020).

Lee et al. (2023) shows that small transformers learn arithmetic when inputs and intermediate steps are tightly structured, with tiny decoder only models mastering addition and multiplication and generalizing to longer lengths under disciplined formats and explicit steps . Follow up work reports algorithmic gains from looping the same parameters across steps (Yang et al., 2024) and from self improvement that moves from easy to hard cases (Lee et al., 2025). Our approach follows the same idea by typing numbers and containers and by constraining the decode, and it can provide trusted in coverage labels for self improvement while routing hard or out of coverage prompts to larger models.

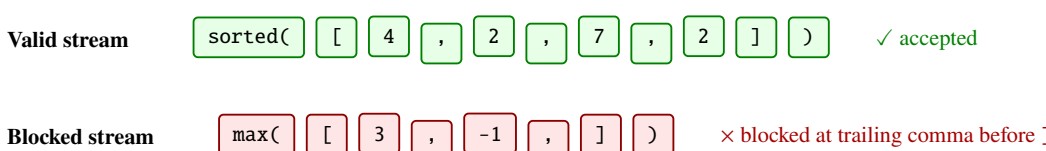

**Valid stream**    `sorted(` `[` `4` `,` `2` `,` `7` `,` `2` `]` `)`    ✓ accepted

**Blocked stream**    `max(` `[` `3` `,` `-1` `,` `]` `)`    ✗ blocked at trailing comma before ]

Figure 3: **Streaming decode with a grammar gate.** The model emits one token at a time and the DFA checks each new token. The top row shows a valid stream for `sorted([4,2,7,2])`. Every token is accepted, so decoding finishes with a correct one line program. The bottom row shows an invalid stream for `max([3,-1,])`. Decoding stops at the trailing comma before the closing bracket because the list automaton allows either another integer or the closing bracket and never a comma right before `]`. Green tokens mark accepted steps and the red token marks the first blocked step. The gate prevents malformed outputs early and hands control to the repair path when needed.

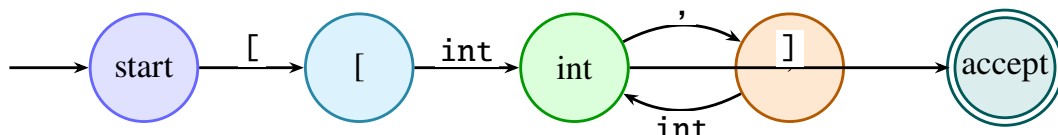

Figure 4: **Minimal DFA for integer lists.** The automaton enforces two simple rules: there must be at least one element and there cannot be a trailing comma. Decoding starts at START, reads `[`, then expects an integer in INT. From INT it can read a comma and loop to INT for another element, or read `]` and move to ACCEPT. Any other token sends the stream to a sink state and decoding stops. As a result it accepts `[int]` and `[int, int, ...]` and rejects `[int, ]`. We use this gate during generation to block malformed lists early and to hand off cleanly to the repair path if needed.

**Datatype tagging ⇒ canonical rewrite ⇒ one-line Python**

> **Prompt** → Subtract 5 from 12.
> **Tagged** → Subtract <INT>5 from <INT>12.
> **Emit** → 12 - 5

> **Prompt** → Arrange 4, 2, 7, 2 in ascending order.
> **Tagged** → Arrange <LIST>[<INT>4, <INT>2, <INT>7, <INT>2<LIST>] in ascending order.
> **Emit** → sorted([4, 2, 7, 2])

> **Prompt** → How many 'r' in 'strawberry'?
> **Tagged** → How many <STR>'r'<STR> in <STR>'strawberry'<STR>?
> **Emit** → 'strawberry'.count('r')

> **Prompt** → Add one 15 times.
> **Tagged** → Add <INT>1 <INT>15 times.
> **Emit** → sum([1]*15)

Figure 5: **Datatype tagging enables canonical rewrites and one-line Python.** Each row shows a natural language prompt, its tagged version, and the emitted single-line Python expression. Tags make numerals and container boundaries explicit, which lets a canonicalizer resolve order-sensitive phrasing and enumerations into safe one-liners.

## 5 LIMITATIONS

Our scope is narrow. We handle one line integer arithmetic and list operations. Extending to floats, richer strings and Unicode, nested or heterogeneous containers, dictionaries, or matrices would need more tags and larger grammars for the DFA and the AST checks, and probably a larger model. Grammar and AST checks ensure the shape of the output rather than full semantics (Scholak et al., 2021; Poesia et al., 2022; Willard & Louf, 2023; Beurer-Kellner et al., 2024).

Security still needs defense in depth. We use time limits and memory limits, caps on input length and list length, and Unicode normalization to reduce confusables and mixed scripts (Unicode Consortium, 2024a;b;c). We run with strict process isolation and we block imports by default, and we harden validators with property based tests (Claessen & Hughes, 2000). Running one line programs inside a WebAssembly runtime gives stronger isolation than running inside the process (Haas et al., 2017). Structure aware decoding lowers error rates but it does not remove arithmetic brittleness or hallucination (Cobbe et al., 2021; Ji et al., 2023). Moreover, any move to floating point must handle well known numerical issues (Goldberg, 1991).

## 6 BROADER IMPLICATIONS AND FUTURE WORK

Typed prompting with grammar first decoding is a general design rather than a niche trick. It maps free form requests into a small and auditable program space and it prunes invalid strings during decoding (Scholak et al., 2021; Poesia et al., 2022; Willard & Louf, 2023; Beurer-Kellner et al., 2024; 2023). The smaller search space improves reliability and cost, and prior work reports large cost reductions and near $2\times$ speedups when constraints are paired with speculative decoding (Beurer-Kellner et al., 2023; 2024). In production this pattern fits cascades and routers. We accept when grammar and AST checks pass and otherwise we abstain and hand off to a larger model (Chen et al., 2023; Ding et al., 2024; Wang et al., 2024; Dohan et al., 2022). Confidence aware early exit and speculative drafting further reduce latency (Schuster et al., 2022; Leviathan et al., 2023). Safety also improves. Grammar gates and AST whitelists narrow prompt injection and unsafe output risks, and sandboxing with Unicode normalization, property based tests, and guardrails adds defense in depth (OWASP, 2023; Unicode Consortium, 2024a;b;c; Claessen & Hughes, 2000; Haas et al., 2017; NVIDIA, 2023; Sheshadri, 2023).

## 7 REPRODUCIBILITY

Here is a short path to reproduce our results. The notebooks are in the Supplementary Material.

**1) Build the data.** Generate gold single line programs for the five skills. Validate each with `ast.parse` and safe execution. Use a stronger model to generate prompts while keeping numerals and list entries unchanged. Save JSON with fields `source`, `skill`, `prompt`, `code`, `tagged_prompt`, `tagged_code`. Make a 90/10 split stratified by skill and list length.

**2) Tag and tokenize.** Insert inline tags for literals and containers on the prompt and on the code. Use `<INT>`, `<FLOAT>`, `<BOOL>`, `<STR>`, `<LIST>`, `<TUPLE>`, and `<|END|>`. Ensure round trip recovery so that detagging returns the original text byte for byte. Register each tag as a single tokenizer token.

**3) Train the model.** Feed `tagged_prompt` `<|END|>` `tagged_code`. Train GPT-2 small with a causal LM loss only on the code tokens. Optionally add a span head over the prompt that predicts a binary mask for digits and container punctuation. Use one to three epochs with learning rate $2 \times 10^{-5}$ and batch size eight or sixteen and maximum sequence length 256. Fix seeds. Save base and auxiliary heads separately to avoid tied weight issues.

**4) Run inference.** Decode under a DFA that enforces the allowed grammar. Stop on EOS or on newline or on `<|END|>`. Detag, then check AST shape. For add and sub require two integer constants. For `max`, `min`, and `sorted` require one list of integer constants. If checks fail apply a deterministic canonicalizer that resolves intent synonyms, normalizes numbers, extracts lists, and fixes operand order. For counting and repeated addition emit `s.count(c)` and `sum([x]*n)` directly. Execute only inside the sandbox.

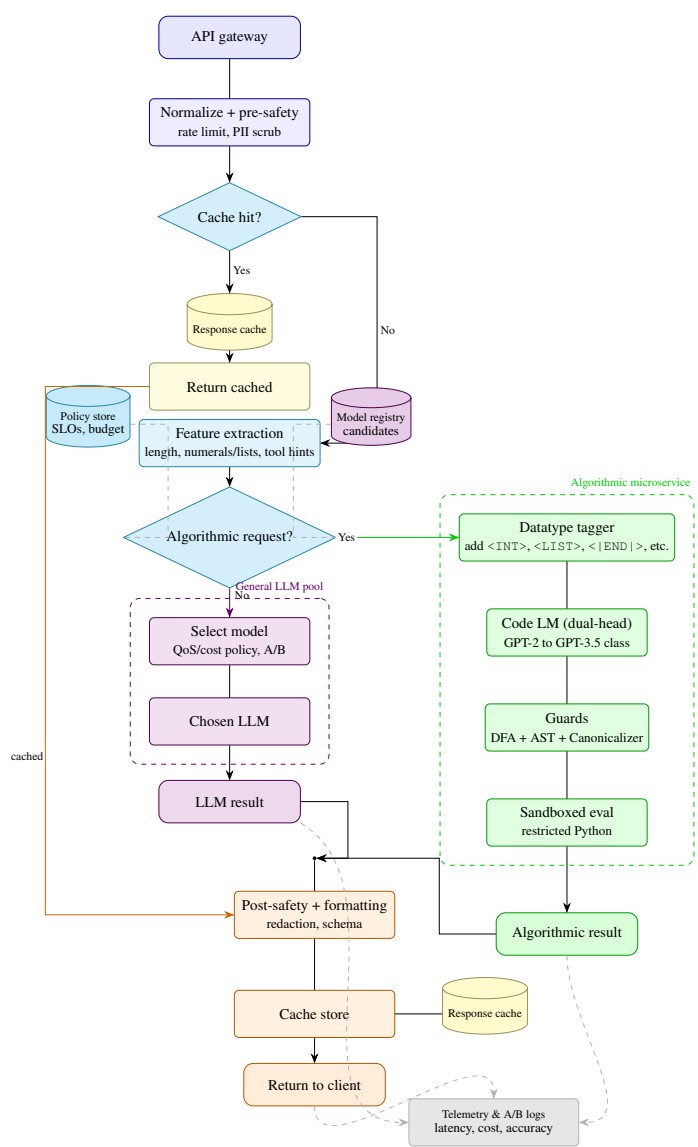

Figure 6: **A proposed industrial routing pattern for LLM systems.** Requests first pass through an API gateway with light pre-safety checks such as rate limiting and PII scrubbing, and the cache is checked so a hit returns immediately. A central router then looks at simple features (for example length, the presence of numerals or lists, and tool hints) and uses a policy store and a model registry to decide whether the request is algorithmic. If it is not, the request follows a general LLM path where a model is chosen by quality, cost, or A/B policy and the prompt is answered directly. If it is, the request goes to an algorithmic microservice that uses a typed, grammar-first pipeline: a datatype tagger, a small dual-head code model, deterministic guards with a DFA and an AST check plus a canonicalizer, and finally sandboxed execution to produce a safe one-line answer. Both paths then join a common post-safety and formatting stage, the cache is updated, and the response is returned to the user while telemetry and A&B logging track latency, accuracy, and cost. This design routes simple algorithmic prompts to a fast and auditable service with strong correctness guarantees, and sends everything else to a general LLM pool, which reduces latency and cost without losing coverage.

LLM USAGE

The authors acknowledge the use of ChatGPT (GPT-5 Pro) for assistance with retrieval and discovery of related work, for drafting text at the paragraph level, and and for helping generate the TikZ figures.

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
