# OpenReview forum: "Datatype tagging and prompt alignment: a recipe for boosting LLMs on algorithmic tasks"
_ICLR.cc/2026/Conference — ICLR 2026 Conference Withdrawn Submission_

### Official Review · Reviewer_FWAP · 2025-10-30

**Soundness:** 1
**Presentation:** 1
**Contribution:** 1
**Rating:** 0
**Confidence:** 4

**Summary:**

This paper evaluates if natural language utterances with tagged data types can be converted one-line Python snippets by a small language model using constrained decoding. During training, a separate model head predicts properties of the predicted tokens with the aim of stabilizing training. The authors claim that constrained decoding improves performance from 0.58 to 0.82—the only two performance metrics mentioned in this paper.

**Strengths:**

The idea of leveraging small models to perform simple tasks can reduce the impact of ML on computational resources.

**Weaknesses:**

This paper only considers a *very* simple domain-specific language and provides no details on how sample problems were generated. Based on the few samples in the paper, I am surprised that a fine-tuned GPT-2 model only achieves 58%, which raises significant concerns about the quality (and size) of the data.

The paper assumes that user prompts are annotated with data types. Again, given the simplicity of the presented tasks, I would not expect these tags to be relevant in the input.

There is very little information on the dataset, such as number of elements or generation process. I would expect most modern LLMs to achieve (near) perfect scores on these simple tasks. I also expect that any sufficiently large and qualitative dataset will cause any modern SLM to be fine-tunable to achieve very high scores—even without constrained decoding.

There are virtually no experiments and results. There's no ablation on the effect of the explicit data types, of the span probe (besides a very vague *stabilizes training*).

**Questions:**

Were any experiments performed to validate the three hypotheses, and what were the results?
What is the size of the dataset?
How do other (baseline) large (and even small) language models perform on this dataset?

---

### Official Review · Reviewer_kBAL · 2025-10-31

**Soundness:** 1
**Presentation:** 2
**Contribution:** 1
**Rating:** 2
**Confidence:** 3

**Summary:**

The paper presents a proof-of-concept technique of using concepts in PL such as type aware tokenization, token tagging, and constraint decoding to aid LLM’s capability of handling small arithmetic tasks. The work achieves this by introducing type-specific tokens into the vocabulary, using a dual-head LM for tagging, and constructing a DFA for constraint decoding. The paper finds that all of these techniques help achieve higher accuracy and even near-zero failure rate on one type of arithmetic task. The paper also presents a theoretical analysis of the constraint decoding.

**Strengths:**

- The paper shows a neat way of combining programming language techniques with LM
- The idea of adding type tags to the prompt is novel

**Weaknesses:**

- The presentation of the paper is somehow confusing, with poorly explained techniques and also many newly coined terms (like “datatype-aware tokenization” and “grammar-first decoding”)
- The evaluation part of the paper is very hand-waiving. Although the paper describes the setup and the findings, the data curation process for both the training dataset and the benchmark is not explained in the paper. Also, the only quantitative analysis of the evaluation result are two numbers (0.58, 0.82), making the claims in the findings somehow vague.
- There is no ablation study in the paper. Constraint decoding is a well known technique for boosting models’ performance. However it is not known from the paper itself whether the improvement comes from constraint decoding or the other two techniques.
- Although the paper is only a PoC on a very simple task, there are no clear way of applying the techniques described in this paper to actual problems since the techniques require modification of the user prompt to include type information, which is known to be very hard since type deduction is very difficult.

**Questions:**

- How are the training dataset and the benchmark curated?
- How is the model’s performance without the constraint decoding?
- How unstable is the model’s training without the probing? Are there qualitative studies showing the probing’s effectiveness?

---

### Official Review · Reviewer_Mn4M · 2025-11-04

**Soundness:** 2
**Presentation:** 3
**Contribution:** 2
**Rating:** 4
**Confidence:** 4

**Summary:**

- This paper addresses the brittleness of Large Language Models (LLMs) on simple algorithmic and arithmetic tasks.

- The method integrates ideas from programming languages (PL) to make even small models reliable on these tasks:
* Inline Datatype Tagging: Making datatypes explicit in the prompt and code
* Fine-tuning a GPT-2-small model with a main code-generation head and an auxiliary "span probe" head (used only during training) to help the model locate important tokens.
* A constrained decoding process that enforces a finite-state grammar (DFA), validates the output's Abstract Syntax Tree (AST) shape, and includes a deterministic "canonicalizer" to repair outputs.

**Strengths:**

- The paper is well written and clearly structured.

- A major strength of the proposed method is achieving high reliability using a very small model (GPT-2-small).

**Weaknesses:**

- The primary weakness is the method's limited scope. The authors are transparent about this - "We handle one line integer arithmetic and list operations". The entire evaluation is based on just five simple skills: add, subtract, max, min, and sort.

- The paper's main quantitative claim (0.58 to 0.82 accuracy) is based on a "stratified held out suite of 60 GPT-40 prompts". An evaluation set of N=60 is exceptionally small and insufficient to make robust claims, even within the paper's narrow scope. This small sample size makes it difficult to know if the results are statistically significant or generalizable.

**Questions:**

- The test prompts were generated by GPT-4o. How robust is the method to diverse, real-world user phrasings that may be ambiguous (e.g., "take 5 from 12" vs. "what is 12 minus 5")?

---

### Official Review · Reviewer_bifH · 2025-11-04

**Soundness:** 2
**Presentation:** 2
**Contribution:** 2
**Rating:** 2
**Confidence:** 2

**Summary:**

The paper proposes a method of using inline type tagging and grammar-constrained decoding to improve code generation performance on small models for the proposed simple one-line Python tasks. They show that fine-tuning GPT-2-small with their method can achieve much better results.

**Strengths:**

Shows that typing and constraint decoding help even for small models like GPT-2-small

**Weaknesses:**

- The experiment is restricted to specific one-line Python settings which are not practical, and it is not clear how to make it work on more practical settings that involve non-regular languages
- Lack of comparison to other simple baselines. For example, how does the model perform with more fine-tuning data? How large does the model need to be to solve this task? Can you try a more recent model like Qwen-3 1.7B instead of GPT-2?

**Questions:**

- Can you show some error analysis? What are some examples of problems that get solved by using the method? And what are some examples of problems that still cannot be solved?
- At line 241, it says "Datatype tags help" and states some advantages of using it. Can you provide quantitative experimental results showing that datatype tags help in the performance?
- At line 252: "With grammar, AST, and the canonicalizer at inference, end accuracy is comparable or slightly higher, so the probe mainly aids stability and diagnostics rather than test time capacity." Can you provide concrete experimental setup and results to back this claim?
- Does the method generalize to cover Python expressions or any real-world programming language expressions and not be restricted like the toy setting proposed here?

---

### Note · Authors · 2025-11-15

I have read and agree with the venue's withdrawal policy on behalf of myself and my co-authors.